# Using Large Language Models to Detect Outcomes in Qualitative Studies of Adolescent Depression

Alison W. Xin
National Institute of Mental Health
Bethesda, MD, USA
al.xin@nih.gov

Dylan M. Nielson
National Institute of Mental Health
Bethesda, MD, USA
dylan.nielson@nih.gov

Karolin Krause
Université Paris Cité
Paris, France
karolin.krause@u-paris.fr

Guilherme Fiorini
University College London
London, United Kingdom
guilherme.fiorini.18@ucl.ac.uk

Nick Midgley
University College London
London, United Kingdom
nicholas.midgley@ucl.ac.uk

Francisco Pereira
National Institute of Mental Health
Bethesda, MD, USA
francisco.pereira@nih.gov

Juan Antonio Lossio-Ventura
National Institute of Mental Health
Bethesda, MD, USA
juan.lossio@nih.gov

## ABSTRACT

Depression treatment studies often focus exclusively on changes in depressive symptoms, such as low mood, anhedonia, or sleep disruption. However, incorporating other outcomes important to those experiencing depression, such as the quality of interpersonal relationships or quality of life, could improve understanding of the impacts of depression and effectiveness of treatment. After analyzing in-depth interviews with adolescents, parents, and therapists, clinicians produced a novel coding framework that covers additional domains of interest that matter to adolescents, such as relationships, functioning, and well-being. In this paper, we examine whether large language model embeddings can be used to classify the outcomes of this framework from annotated interviews. We compare the suitability of four language models across three different segmentations of interview transcripts, such as conversation turns or non-interviewer utterances. The level of performance achieved by our models makes them useful for a variety of applications, ranging from aiding human annotation of text transcripts to quantifying the presence of outcomes for downstream uses, such as estimating treatment effects or building prognostic models.

## CCS CONCEPTS

• **Applied computing** → **Health informatics**; • **Computing methodologies** → **Information extraction**.

## KEYWORDS

Large language models, BERT, Llama 2, Llama 3, adolescent depression, depression outcomes, mental health

**ACM Reference Format:**
Alison W. Xin, Dylan M. Nielson, Karolin Krause, Guilherme Fiorini, Nick Midgley, Francisco Pereira, and Juan Antonio Lossio-Ventura. 2024. Using Large Language Models to Detect Outcomes in Qualitative Studies of Adolescent Depression. In *Proceedings of Artificial Intelligence and Data Science for Healthcare: Bridging Data-Centric AI and People-Centric Healthcare (AIDSH-KDD '24)*. ACM, New York, NY, USA, 12 pages. https://doi.org/XXXXXXX.XXXXXXX

## 1 INTRODUCTION

Globally, in adolescents aged 10-19 years, the prevalence of major depressive disorder and dysthymia is estimated at 8% and 4%, respectively [1]. Advancing understanding of treatment outcomes is critical in addressing this public health problem. In clinical trials and routine specialist care, around 40% of youth leave treatment without showing meaningful improvement in depressive symptoms, which include low mood, anhedonia, sleep disruption, suicidality, or irritability, defined by the DSM and ICD-11. Less is known about the impact of treatment on other outcomes, such as relationships or quality of life. Between 2007 and 2017, a systematic review of clinical studies of depression found that 94% of studies measured depressive symptoms, 52% measured general functioning, and less than 10% measured any other outcome [2].

Previously, clinical researchers performed a post-hoc analysis of interview transcript data from the qualitative study IMPACT-My Experience (IMPACT-ME [3]), a substudy nested within the Improving Mood with Psychoanalytic and Cognitive Therapies (IMPACT) study of the psychological treatment of adolescent depression [4, 5]. Using qualitative content analysis, they produced a systematic and comprehensive framework of adolescent depression treatment outcomes, identifying seven broad outcome domains, and twenty-nine specific outcomes of interest [6]. Analysis of these outcomes in

qualitative data could complement traditional quantitative measurement of symptom change, providing a more holistic impression of how treatment affects depression.

However, manual qualitative analyses of large volumes of qualitative data is time-intensive and may not always be feasible. Recent developments in natural language processing (NLP), particularly improvements in large language models (LLMs), can help address this challenge by automating the analysis of large volumes of data. A recent survey demonstrated that NLP enables automated screening for symptoms of several mental disorders from text data [7], though most of these studies only address a single binary classification (e.g., depressed and non-depressed) or regression (e.g., severity). Additional limitations are the use of social media data, which complicates clinical integration, and lack of work focused on adolescents.

We aim to train models capable of reliably detecting mentions of fine-grained depression outcomes in the IMPACT-ME study data. We compared several LLMs to investigate whether the use of different embeddings improved detection performance. We use open-source LLMs deployable within our own servers, which reduces concerns with protected health information or personally identifiable information. In this paper, we demonstrate the feasibility of using LLM embeddings as part of models for detecting outcomes across a range of high-level domains. We compare the performance of embeddings from various LLMs, including the recently released Llama 3. Overall, we find that LLM embeddings allow for effective classification of these outcomes, and could be useful for future work on understanding the holistic experience of depression and its treatment.

## 2 RELATED WORK

Various natural language processing techniques can detect mental health disorders and symptoms by automating the analysis of large volumes of data, as described in a recent survey of nearly 400 articles [7]. Social media posts are the predominant data source (81%) [8], followed by interviews (7%), EHRs (6%), screening surveys (4%), and narrative writing (2%) [7]. NLP transforms text into numerical representations, which may include specific linguistic features, language representation features, and others. NLP uses both traditional machine learning (ML) and deep learning-based methods for tasks related to depression, such as risk assessment, symptom detection, and more.

Traditional ML methods usually extract handcrafted features in model training for classification or prediction tasks. Features include linguistic, statistical, and domain-specific features. For example, linguistic features identified by the Linguistic Inquiry and Word Count (LIWC) [9, 10] tool have proven effective in detecting depressive moods and other mental health indicators from language [11–15]. Part-of-speech (POS) tagging [16–18] or extraction of sentiments, emotions, topics, word usage, grammar, and readability have also been used [14, 19–21]. Statistical features also include bag-of-words (BoW) [16, 22], n-grams [23–25], term frequency-inverse document frequency (TF-IDF) [26], and sentence or passage length [27, 28]. Domain-specific features might involve ontologies and dictionaries, such as UMLS [29] or other specialized vocabularies

[16, 30, 31]. Many traditional ML algorithms, such as support vector machines (SVMs) [12, 13, 32, 33], decision trees [34], random forests [34], adaptive boosting [35], k-nearest neighbors (KNN) [36, 37], and logistic regression [13, 38–40], have been applied for depression-related tasks.

Deep learning-based methods garner significant attention due to their superior performance compared to traditional ML methods [7, 41]. In particular, LLMs have become foundational tools for transforming text inputs into quantitative vector representations, or *embeddings*. In contrast to traditional ML, embeddings are learned from data using various algorithms, such as neural networks, rather than being defined by human experts. These embeddings can then be used as inputs for classification models to predict annotations, such as the presence of specific depression markers. Various embedding techniques, including GloVe [42], word2vec [43], and transformer-based models like BERT [44] and RoBERTa [45], effectively identify depression markers in text [25, 46]. Deep learning methods are generally categorized into convolutional neural network (CNN)-based, recurrent neural network (RNN)-based, and transformer-based approaches [7, 47]. CNN architectures incorporate convolutional, pooling, and fully connected layers [48, 49]. RNN architectures, such as long short-term memory (LSTM) and gated recurrent unit (GRU), often incorporate attention mechanisms and hierarchical attention networks for multi-level semantic information extraction, making them well-suited for sequential data like text[13, 50–53]. Transformer-based methods, including BERT, RoBERTa, Llama [54–56], Mistral [57], and the GPT* series [58], incorporate an attention mechanism that manages long-range dependencies, which are crucial in NLP applications. Transformers can be finetuned for various prediction and classification tasks, and large-scale pre-training improves performance, as demonstrated in specialized domains such as depression detection [59–64].

Most previous studies tackle broad binary classification problems (i.e., depression and control group). Additionally, the lack of interpretability in many models prevents clinicians from relying on the outcomes of automated screening techniques. Therefore, the scientific community has initiated several efforts to improve the clinical applicability of machine learning studies, including the Early Risk Prediction on the Internet (eRisk) workshop, which has been part of the Conference Labs of the Evaluation Forum (CLEF) since 2017. eRisk provides a collaborative environment for developing methods and practical approaches for early detection of health risks on the Internet, including depression. In 2023, eRisk featured a depression-related task (Task 1) [65] that involved ranking sentences based on their relevance to each of the 21 symptoms of depression derived from the Beck Depression Inventory–II (BDI-II) [66]. Symptoms included pessimism, thoughts about suicide, or sleep problems, rated on a severity scale from 0 to 3. Outside of eRisk, other studies aggregate symptoms from different questionnaires, such as the BDI-II [67, 68] and PHQ-9 [69], and transformer-based models, such as BERT, can screen for depression in patients [70].

However, these initiatives mainly rely on social media data, which limits clinical integration due to issues with standardization and reliability. Additionally, limited research focuses on adolescent participants, highlighting the need for studies that address the unique factors affecting depression detection in this age group. Our work differs in that it focuses only on the detection of symptoms

particularly relevant to adolescents and uses a psychiatric clinical dataset rather than social media. In addition, given the sensitivity of the dataset, our approach was developed using the latest open-source large language models, such as Llama, rather than commercial ones such as GPT or Claude.

# 3 MATERIALS AND METHODS

## 3.1 Data

*3.1.1 IMPACT-ME interviews.* Interviews were taken from IMPACT-My Experience (IMPACT-ME) [6], a qualitative study within the Improving Mood with Psychoanalytic and Cognitive Therapies (IMPACT) trial [4]. IMPACT examined the efficacy of Brief Psychosocial Intervention (BPI), Cognitive Behavioral Therapy (CBT), and Short-Term Psychoanalytic Psychotherapy (STPP) for adolescents aged 11-17 diagnosed with unipolar Major Depressive Disorder. In IMPACT-ME, interviews were conducted with adolescent patients, parents, and therapists at treatment start, end, and one-year follow-up, exploring therapy experiences and observed changes [6].

*3.1.2 Qualitative analysis and annotation.* Krause et al. conducted a secondary qualitative analysis on these interviews to explore the range of treatment outcomes relevant to patients. Interviews from the end of treatment were transcribed verbatim and included pauses, filler words, interruptions, and typos. Participants were excluded if any of the three interviews were missing, if treatment ended within the first three sessions, or if they were referred to inpatient care. Of the remaining 34 cases (9 BPI, 9 CBT, and 16 STPP participants; 102 interviews), the average age was 16.2 years ($s$ = 1.5, range = 12-19), and 21 (61%) were female. To categorize outcomes, Krause et al. first designed an a priori coding framework based on existing taxonomies of treatment outcomes. During annotation, outcome-relevant passages were extracted, and the coding framework was further modified to incorporate new themes. The final framework contained 29 specific outcome categories within seven high-level domains [6], listed in Table 1 and described further in Table 4 in Appendix A. All annotations were performed by one researcher.

*3.1.3 Dataset splitting.* We split the dataset of 34 subject cases into a training set of 26 subjects and a test (holdout) set of 8 subjects. Transcripts were grouped by subject (i.e., by triplets of interviews relating to an adolescent participant) to prevent training and teseting models on data from the same person. Test cases were determined by manually balancing positive and negative examples for all specific outcomes. The test set was not used in this paper and is reserved for future evaluation.

## 3.2 Preprocessing

*3.2.1 Conversion to labeled text blocks.* Empty lines and header information, such as subject ID and interviewer ID, were removed from transcription files before analysis. Transcripts were split into speaker blocks, which were marked by the start of a new paragraph in the transcript. The original IMPACT-ME annotations created by Krause et al. were produced by highlighting excerpts of the transcript relevant to a specific outcome. These excerpts could start or end at any position in a speaker block, and a block could contain multiple annotations. For our models, an entire text block was

**Table 1: Names of the high-level domains and the specific outcomes found during qualitative analysis [6].**

| Domain | Specific outcomes |
|---|---|
| (A) Symptom change | (1) Mood and affect, (2) anger and aggression, (3) appetite, (4) sleeping and energy, (5) self-harm, (6) suicidality, (7) anxiety, (8) other comorbidities |
| (B) Coping and self-management | (1) Behavioral activation, (2) coping and resilience, (3) cognition and behavior |
| (C) Functioning | (1) Global functioning, (2) executive functioning, (3) academic and vocational functioning, (4) social functioning |
| (D) Personal growth | (1) Assertiveness, (2) autonomy and responsibility, (3) identity, (4) processing past and present, (5) confidence and self-esteem, (6) feeling seen and seeing differently |
| (E) Relationships | (1) Ability to talk, (2) family functioning and relationships, (3) friendships, (4) peer relationships, (5) romantic relationships |
| (F) Wellbeing | (1) Peace of mind, (2) optimism, (3) future orientation |
| (G) Parental support and wellbeing | (1) Parental support, (2) parental wellbeing |

labeled as positive if any proportion of it contained text flagged as positive for an outcome. Blocks were labeled for the presence of 31 specific outcomes, 7 domains (each containing a disjoint subset of the outcomes), and presence of any positive label, totaling 39 binary label indicators for every text block (details in Appendix A Table 4). The number of positive samples for each label can be found in in Appendix A Table 5.

*3.2.2 Transcript segmentations.* The *Original* segmentation of text generated from annotations, containing 32,520 blocks, included various uninformative text segments. Outcome-relevant dialogue would often be interspersed with interjections, acknowledgments, or requests for elaboration, e.g., an interviewer saying "okay" or "yes" to encourage a patient would be included within the excerpt and labeled as positive in our dataset. To address these uninformative text blocks, we created two additional segmentations of the transcript, Monologue and Turns, described below. An example contrasting these segmentations with the Original segmentation can be found in Appendix A Table 3.

*Monologue*: We discarded all interviewer speech and blocks with twelve or fewer characters. We manually determined the cutoff by examination of the labeled text in the training set. By only retaining non-trivial interviewee text, we aimed to produce "monologues" about the study experience, although some interviews, such as those conducted jointly with both parents of a patient, retained multiple interviewees interacting in dialogue. Of the original 32,520 blocks, 12,941 were retained in this filtration.

*Turns*: We partitioned blocks at each interview utterance, grouping together sequential pairs of utterances by interviewer and interviewee into "turns" of the conversation. By concatenating blocks, the Turns segmentation kept informative interviewer questions together with short interviewee responses that were otherwise uninformative (e.g., "I: How has your mood been?" "P: Fine..."). For interviews with multiple interviewees, all utterances between interviewer utterances were concatenated into the same turn. This process produced 16,139 blocks of text.

**Table 2: Maximum sequence length (max. seq.), hidden dimension size (hidden dim.) and millions of parameters (params) for the LLMs used to generate embeddings.**

| Model | Max. seq. | Hidden dim. | Params ($10^6$) |
|---|---|---|---|
| BERT | 512 | 768 | 110 |
| MentalBERT | 512 | 768 | 110 |
| MentalLongformer | 4096 | 768 | 102 |
| Llama 2-7B | 4096 | 4096 | 7,000 |
| Llama 3-8B | 8192 | 4096 | 8,000 |

The training set contained 25,852 blocks in the Original, 10,008 blocks in Monologue, and 12,814 blocks in Turns. Full counts of the number of positive examples for each label in the complete and training set can be found in Appendix A Table 5.

## 3.3 Methods

*3.3.1 Large language model embeddings.* Embeddings were produced with various transformer-based LLMs. As a baseline, we used the base variant of BERT [71], a common choice for various NLP tasks, such as sentiment analysis or summary generation. Additionally, we included MentalBERT, a BERT model pretrained on additional data collected from various Reddit communities related to mental health discussion [72]. Because of the long passages present in all segmentations, we also included MentalLongformer, a derivation of Longformer [73] pretrained on the same mental health data as MentalBERT [74]. Furthermore, we included Llama 2-7B, and Llama 3-8B, state-of-the-art open source models [54, 56].

LLMs produced embeddings of size $b \times l \times d$, where $b$ is the batch size, $l$ is the sequence length, and $d$ is the hidden dimension of the model. For each model, we used $b = 1$, i,.e., passing individual blocks to the LLM. Sequence length (number of tokens), varied based on a passage's length and the model-specific tokenizer. Passages exceedeing a model's maximum sequence length were truncated before embedding. Embeddings were averaged across all tokens in the sequence to produce a $d$-dimensional vector of predictors for each text block. An overview of model details can be found in Table 2.

*3.3.2 Training classification models.* To classify labels, we trained L2-penalized logistic regression models on the $d$-dimensional averaged embedding vector for each passage. Models were trained and evaluated with a 4-fold cross-validation (CV) loop. For each of the 4 test folds, the $C$ hyperparameter of logistic regression was tuned with inner 3-fold CV, using the same fold partitions as the outer 4-fold CV. Data were grouped by subject ID and stratified by label. Models for labels A8, E4, and E5 could not be trained because fewer than four subjects were present in the development data. To adjust the loss function for the imbalance between positive and negative examples in every label, errors in positive examples were multiplied by the ratio of positive to negative examples in that label.

## 4 RESULTS

### 4.1 Classification performance for each label

Our first goal was to investigate classification performance of each of the embedding models for our 39 binary labels (31 specific outcomes[1], 7 high-level domains, and presence of any outcome). For each model, we computed the area under the ROC curve (ROC AUC) for each test fold and reported the average ROC AUC across folds [75]. Classification performance fell within 0.6-0.9 for the Original segmentation and 0.7-1.0 for the Monologue and Turns segmentations, seen in Figure 1. (details in Table 6 in Appendix B).

In the Original segmentation, D1 performs the best across all models. In Monologue, the best performer was one of D3 or F2. In Turns, A3 and D3 performed well for all models, but the top performer for MentalLongformer was F2. For any combination of model and segmentation, the lowest performer tended to be D2 or G1, with A2 performing poorly in Original. Many labels were inconsistent across models and segmentations. For example, A5 was in the top four for all models in the Original segmentation, but underperformed in Monologue and Turns in non-Llama models. The worst classified labels tended to have high variance in performance across embedding models, though the relative rankings are consistent. For every embedding, the averaged ROC AUC for models of "Any" outcome were between 0.75-0.85. Further details on relative classification performances can be found in Table 7 in Appendix B.

### 4.2 Statistical comparison of embedding models

Our second goal was to determine whether embeddings from a particular large language model had consistently better performance than others. For our 28 specific outcomes, we tested the null hypothesis of no difference in model performance with the Friedman test [76]. We excluded aggregate labels, i.e., the seven domain labels and the "Any outcome" label, to avoid double-counting. Friedman test results were $Q_3 = 8.571, p = 0.0356$ for Original; $Q_3 = 13.16, p = 0.00431$ for Monologue; and $Q_3 = 12.56, p = 0.00570$ for Turns. At significance level $\alpha = 0.05$, the Friedman test results supported rejection of the null hypothesis of model equivalence, and we proceeded with the post hoc Bayesian comparison tests [77].

The Bayesian post hoc test indicated that both Llama models had probability $\geq 0.94$ of outperforming any other non-Llama model (Figure 2). BERT had a $< 0.04$ probability of outperforming any model except for MentalLongformer, where the probability of BERT being better was 0.14, 0.15 and 0.08 for Original, Monologue, and Turns, respectively. Llama 2-7B and Llama 3-8B had a 0.84, 0.71 and 0.92 probability of practical equivalence for Original, Monologue, and Turns. MentalLongformer and MentalBERT as well as BERT and MentalBERT had practical equivalence probabilities of 0.19 to 0.38 in Original and Turns. All other pairwise model comparisons returned $\leq 0.06$ probability of practical equivalence.

## 5 DISCUSSION AND CONCLUSION

Models generally performed well, even for labels with very few positive examples. Across the 36 labels considered, performance was

---

[1]Results are reported for 28/31 specific outcome labels (i.e., 36/39 binary labels), as three labels did not contain enough subjects for 4-fold CV.

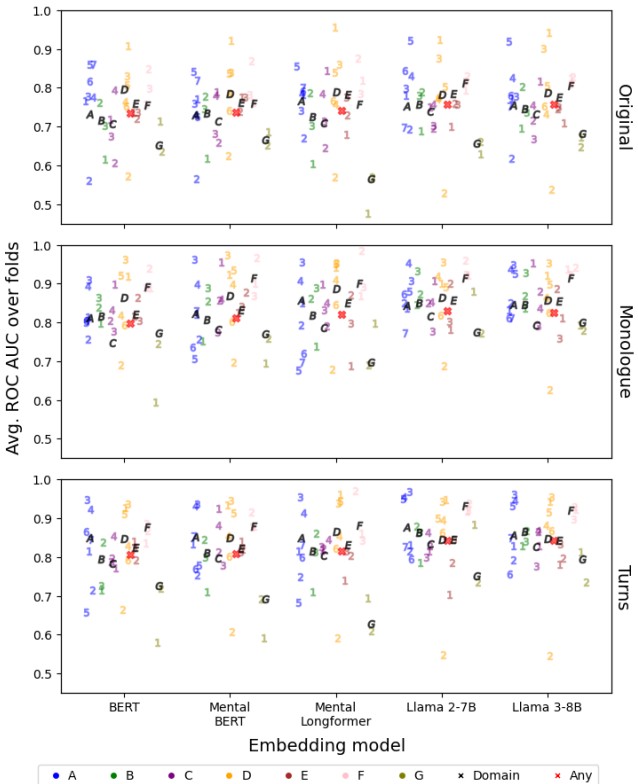

**Figure 1: Average ROC AUC performance for logistic regression models, with horizontal jitter for clarity. Results for each label are grouped vertically by domain (labeled by color) and horizontally by segmentation (labeled on right axis). Domains are "Symptom Change" (A), "Coping and self-management" (B), "Functioning" (C), "Personal growth" (D), "Relationships" (E), "Peace of mind" (F), and "Parental support and wellbeing" (G). Numbers indicate the performance of each specific outcome within a domain, black letters the domains, and the red X represents "Any" of the outcomes.**

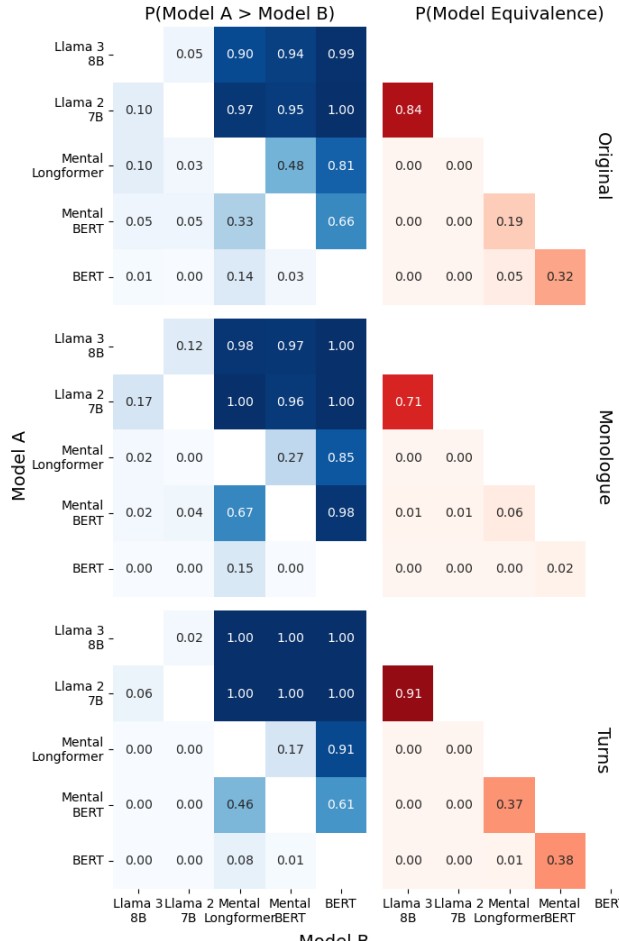

**Figure 2: Bayesian model comparison test, with a region of practical equivalence of 0.01. Results for each segmentation are grouped by row. (Left, blue) Probability that Model A (y-axis) outperforms Model B (x-axis). (Right, red) Probability of Model A and Model B being practically equivalent.**

never below an average ROC AUC of 0.60 for every model within a segmentation. Given these results, we believe that classifiers using LLM embeddings as inputs could prove useful for detecting fine-grained outcomes. On the other hand, it is unclear why specific labels were easier or harder to classify. Even within the same domain, specific outcomes can run a wide range, e.g., D3 being best overall while D2 is worst overall. The relative performances of aggregated labels, such as labels for high-level domains A through G, tend to be consistent across models within a segmentation, suggesting that some variability may be due to the small number of positive examples.

The Bayesian comparison test between models suggests that, of the models investigated, Llama models produce more informative embeddings for classification. The test also suggests that Llama 2-7B and Llama 3-8B have a high probability of practical equivalence, which is unsurprising considering their architectural similarities.

The non-negligible probability of practical equivalence for MentalBERT and MentalLongformer is also unsurprising, considering the models are pre-trained with the same set of mental health data. MentalBERT, though fine-tuned on domain-specific data, only has a high (0.98) probability of outperforming BERT on Monologue, and has a moderate probability of practical equivalence (0.32, 0.38) on Original and Turns. Additionally, although the Bayesian comparison test suggests that Llama models have high probabilities of outperforming the other models tested, the advantage is not very large. Other concerns, such as resource usage, may be a deciding factor in choosing an embedding model for different tasks. Llama 2-7B and Llama 3-8B, for example, require a GPU to perform inference, while BERT, MentalBERT, and MentalLongformer can produce embeddings using the CPU available in a standard laptop.

Based on our results, our methodology may be useful for similar datasets or labels at a comparable level of granularity. However, it

is unclear how reported performance could reflect idiosyncrasies of this dataset. The small sample size, both in total positive labels and number of subjects, may prove an obstacle when applying these particular models to other datasets. For example, the adherence to transcribing the exact utterances of the interviews is not common in written text or in machine transcription, which often removes pauses, filler words, and accent indicators.

The $k$-fold cross-validation results should provide reasonable estimates of model performance in new participants, given that we have reported on all the experiments that we have carried out. Nevertheless, the final analysis of generalization, and model variance, should be performed on the test set, which we are currently withholding to allow for further model development on the training set. When testing on the holdout, we will produce performance estimates for labels for which models could not be trained and tuned during the $k$-fold cross-validation step due to sparsity.

Future model development will work on two fronts. The first is to improve text representation, either through more advanced models or by fine-tuning LLMs for these prediction tasks. The second will be to improve the generalization of prediction models through, for example, supplementing the training data with additional synthetic examples, paraphrased by an LLM from existing ones.

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

## A ANNOTATION DETAILS

Table 3 illustrates how segmentations might be created from a passage, though the text example lacks many of the interview transcripts' idiosyncrasies, such as inclusion of hesitations and filler words. Additionally, the example Monologue segmentation is more coherent than the true dataset, as realistic answers are often difficult to understand without the context of the interview question.

Table 4 provides more description of the specific outcomes of interest in the IMPACT-ME interviews. The addendum to Krause et al. also contains examples of specific outcomes to illustrate how they may appears in the transcript [6].

**Table 3: A comparison between how blocks would be formed between the Original, Monologue, and Turns segmentation. A change in text color indicates the boundary of the input block. The example text is not based on any interview in the dataset.**

| Original | Monologue | Turns |
|---|---|---|
| I: How are you? | | I: How are you? |
| P: Ok... | | P: Ok... |
| I: Just ok? | | I: Just ok? |
| P: Not feeling the best about school. | P: Not feeling the best about school. | P: Not feeling the best about school. |
| I: Why? | | I: Why? |
| P: I hate this group project. | P: I hate this group project. | P: I hate this group project. |
| I: Mmhmm. | | I: Mmhmm. |
| P: They always ignore me so I have to work alone. | P: They always ignore me so I have to work alone. | P: They always ignore me so I have to work alone. |

## B MODEL PERFORMANCE DETAILS

ROC AUCs reported in Table 6 are averaged across each of the $k$ outer folds, with $k = 4$. By sorting the labels by average overall rankings (Table 7), we can observe that around half of the specific outcomes show wide variance in the relative performance for each model and segmentation.

**Table 4: Names of the labels and a brief description. Unless otherwise indicated, assume descriptions refer to changes with the adolescent patient.**

| Abbrev. | | Name | Description |
|---|---|---|---|
| Any | | | $A \cup \cdots \cup G$ |
| A | | Symptom change | $A_1 \cup \cdots \cup A_8$ |
| | A1 | Mood and affect | Less low and depressed; low mood is more fleeting, less overwhelming. |
| | A2 | Anger and aggression | Less angry, irritable, aggressive; fewer outbursts; better able to manage temper. |
| | A3 | Appetite | Healthier appetite and weight. |
| | A4 | Sleeping and energy | Healthier sleep patterns and energy levels. |
| | A5 | Self-harm | Less self-harm (e.g., cutting, trichotillomania) |
| | A6 | Suicidality | Reduced suicidal ideation and behavior |
| | A7 | Anxiety | Fewer fears, worries, panic attacks; less social anxiety; engaging in activities |
| | A8 | Other comborbities | E.g., substance abuse or obsessive-compulsive symptoms |
| B | | Coping and self-management | $B_1 \cup B_2 \cup B_3$ |
| | B1 | Behavioral activation | More active; returning to hobbies or engaging in new activites; sense of purpose, routine, and structure |
| | B2 | Coping and resilience | Specific coping strategies, understanding of feelings, thoughts, and behaviors; anticipating and managing challenges; more resilient, greater self-efficacy, sense of control |
| | B3 | Cognition and behavior | Challenges negative automatic thoughts, more flexible thinking styles |
| C | | Functioning | $C_1 \cup C_2 \cup C_3 \cup C_4$ |
| | C1 | Global functioning | Better function across range of life domains, engages in typical adolescent activities |
| | C2 | Executive functioning | Able to get things done; improved concentration, motivation, planning, organization |
| | C3 | Academic and vocational functioning | Attends school more regularly; works more effectively in school, achieves better results |
| | C4 | Social functioning | More outgoing and talkative, more present within friendship groups, more socially connected; easier to make conversation, relate to others, be mindful of others' feelings |
| D | | Personal growth | $D_1 \cup \cdots \cup D_6$ |
| | D1 | Assertiveness | Better able to stand up for needs and opinions, overcome urge to please, can express disagreement or disapproval when appropriate |
| | D2 | Autonomy and responsibility | More independent, takes responsibility for life and actions |
| | D3 | Identity | Finding out who they are and how to be themselves around other people; less idealised self-images that can accommodate both positive and challenging personality traits; positive and negative feelings |
| | D4 | Processing past and present | Making sense of challenging past or ongoing experiences such as bereavement, parental divorce, or family conflict |
| | D5 | Confidence and self-esteem | More confident, less insecure, less vulnerable to judgement, higher self-regard |
| | D6 | Feeling seen and seeing differently | Feeling listened to, understood, or cared for; experiences of being worth of another's attention; new perspectives; opportunity to release feelings, thoughts or memories |
| E | | Relationships | $E_1 \cup \cdots \cup E_5$ |
| | E1 | Ability to talk | More able to talk about feelings and thoughts, which helps deepen relationships; stronger support network facilitates opening up |
| | E2 | Family functioning and relationships | Getting on better with their family: less conflict, better understanding from family; easing of entrenched tensions between family members; families communicate more openly; role within the family system clarified |
| | E3 | Friendships | Reactivation or deepening of existing friendships, expanding friendship groups or changing friends by turning towards more supportive friendships |
| | E4 | Peer relationships | Getting on better with peers in school |
| | E5 | Romantic relationships | Getting on better with romantic partner |
| F | | Wellbeing | $F_1 \cup F_2 \cup F_3$ |
| | F1 | Peace of mind | Calmer, more balanced, relaxed, and carefree; feeling as if a weight had been lifted off their shoulders; more accepting of things that cannot be changed |
| | F2 | Optimism | More positive and optimistic outlook into their lives and the future |
| | F3 | Future orientation | Can make plans for the future and have goals |
| G | | Parental support and wellbeing | $G_1 \cup G_2$ |
| | G1 | Parental support | Parents are better able to understand their child's difficulties and more aware of how their parenting practices may contribute to these difficulties; parents learn to support and parent their child more effectively |
| | G2 | Parental wellbeing | Parents feel less guilty, isolated, stressed, and worried; parents feel reassured, supported, and able to express their own frustrations and issues |

**Table 5: Counts of labels of interest within the entire dataset and the training dataset, sorted by segmentation.**

| Label | All data | | | Training | | |
|---|---|---|---|---|---|---|
| | Original | Monologue | Turns | Original | Monologue | Turns |
| Any | 2116 | 1042 | 1166 | 1543 | 732 | 840 |
| A | 624 | 313 | 345 | 414 | 197 | 225 |
| A1 | 307 | 164 | 179 | 223 | 117 | 130 |
| A2 | 142 | 63 | 75 | 107 | 43 | 55 |
| A3 | 42 | 24 | 24 | 18 | 10 | 10 |
| A4 | 82 | 42 | 43 | 64 | 31 | 32 |
| A5 | 42 | 23 | 25 | 18 | 10 | 11 |
| A6 | 36 | 19 | 21 | 23 | 13 | 14 |
| A7 | 63 | 29 | 35 | 35 | 13 | 18 |
| A8 | 12 | 6 | 7 | 7 | 3 | 4 |
| B | 592 | 275 | 324 | 436 | 191 | 234 |
| B1 | 112 | 52 | 60 | 81 | 33 | 41 |
| B2 | 432 | 204 | 236 | 311 | 140 | 167 |
| B3 | 119 | 48 | 68 | 106 | 42 | 61 |
| C | 404 | 213 | 231 | 351 | 181 | 198 |
| C1 | 19 | 12 | 13 | 12 | 8 | 9 |
| C2 | 87 | 45 | 51 | 80 | 41 | 46 |
| C3 | 206 | 111 | 121 | 180 | 94 | 104 |
| C4 | 144 | 80 | 84 | 124 | 68 | 72 |
| D | 450 | 255 | 267 | 312 | 177 | 188 |
| D1 | 37 | 25 | 25 | 28 | 20 | 20 |
| D2 | 84 | 43 | 44 | 61 | 31 | 32 |
| D3 | 55 | 38 | 39 | 49 | 34 | 35 |
| D4 | 57 | 35 | 35 | 29 | 17 | 17 |
| D5 | 144 | 79 | 83 | 83 | 47 | 50 |
| D6 | 119 | 62 | 68 | 98 | 50 | 56 |
| E | 441 | 228 | 255 | 287 | 147 | 168 |
| E1 | 43 | 27 | 28 | 26 | 15 | 16 |
| E2 | 273 | 139 | 155 | 151 | 77 | 89 |
| E3 | 130 | 69 | 76 | 113 | 60 | 65 |
| E4 | 34 | 15 | 19 | 25 | 11 | 14 |
| E5 | 9 | 2 | 7 | 9 | 2 | 7 |
| F | 134 | 79 | 85 | 109 | 62 | 68 |
| F1 | 48 | 26 | 30 | 36 | 19 | 23 |
| F2 | 24 | 16 | 16 | 22 | 15 | 15 |
| F3 | 65 | 40 | 42 | 53 | 30 | 32 |
| G | 123 | 60 | 66 | 97 | 46 | 51 |
| G1 | 21 | 13 | 13 | 18 | 11 | 11 |
| G2 | 106 | 51 | 57 | 82 | 38 | 43 |

**Table 6: Averaged ROC AUC across outer $k$-fold cross-validation. Abbreviations are as follows: Sgmnt.: segmentation, MBERT: MentalBERT, MLong: MentalLongformer, L2-7B: Llama 2-7B, L3-8B: Llama 3-8B.**

| Sgmnt. Model Label | Original BERT | MBERT | MLong | L2-7B | L3-8B | Monologue BERT | MBERT | MLong | L2-7B | L3-8B | Turns BERT | MBERT | MLong | L2-7B | L3-8B |
|---|---|---|---|---|---|---|---|---|---|---|---|---|---|---|---|
| Any | 0.735 | 0.737 | 0.742 | 0.758 | 0.757 | 0.799 | 0.811 | 0.822 | 0.829 | 0.826 | 0.806 | 0.809 | 0.817 | 0.844 | 0.843 |
| A | 0.732 | 0.730 | 0.766 | 0.753 | 0.755 | 0.811 | 0.822 | 0.856 | 0.844 | 0.846 | 0.850 | 0.852 | 0.851 | 0.876 | 0.858 |
| A1 | 0.767 | 0.772 | 0.782 | 0.780 | 0.773 | 0.815 | 0.820 | 0.851 | 0.836 | 0.831 | 0.815 | 0.834 | 0.815 | 0.832 | 0.830 |
| A2 | 0.562 | 0.566 | 0.671 | 0.695 | 0.618 | 0.757 | 0.758 | 0.794 | 0.851 | 0.849 | 0.715 | 0.754 | 0.750 | 0.815 | 0.792 |
| A3 | 0.777 | 0.760 | 0.741 | 0.798 | 0.779 | 0.910 | 0.962 | 0.956 | 0.909 | 0.950 | 0.947 | 0.937 | 0.955 | 0.968 | 0.961 |
| A4 | 0.777 | 0.731 | 0.785 | 0.831 | 0.827 | 0.895 | 0.897 | 0.908 | 0.954 | 0.937 | 0.923 | 0.932 | 0.914 | 0.955 | 0.945 |
| A5 | 0.861 | 0.843 | 0.856 | 0.923 | 0.919 | 0.801 | 0.707 | 0.678 | 0.880 | 0.925 | 0.657 | 0.780 | 0.684 | 0.949 | 0.932 |
| A6 | 0.818 | 0.725 | 0.791 | 0.845 | 0.780 | 0.806 | 0.736 | 0.721 | 0.873 | 0.812 | 0.866 | 0.771 | 0.800 | 0.794 | 0.755 |
| A7 | 0.861 | 0.819 | 0.802 | 0.699 | 0.752 | 0.804 | 0.829 | 0.696 | 0.773 | 0.812 | 0.846 | 0.854 | 0.831 | 0.827 | 0.851 |
| B | 0.717 | 0.734 | 0.726 | 0.757 | 0.746 | 0.816 | 0.808 | 0.819 | 0.850 | 0.846 | 0.795 | 0.811 | 0.816 | 0.864 | 0.866 |
| B1 | 0.615 | 0.619 | 0.604 | 0.690 | 0.696 | 0.798 | 0.752 | 0.739 | 0.846 | 0.836 | 0.715 | 0.711 | 0.710 | 0.786 | 0.829 |
| B2 | 0.762 | 0.777 | 0.786 | 0.790 | 0.802 | 0.840 | 0.854 | 0.861 | 0.873 | 0.869 | 0.839 | 0.844 | 0.860 | 0.879 | 0.868 |
| B3 | 0.702 | 0.746 | 0.703 | 0.780 | 0.738 | 0.863 | 0.890 | 0.890 | 0.933 | 0.929 | 0.727 | 0.800 | 0.814 | 0.864 | 0.842 |
| C | 0.708 | 0.714 | 0.710 | 0.739 | 0.733 | 0.749 | 0.782 | 0.787 | 0.816 | 0.793 | 0.784 | 0.798 | 0.804 | 0.834 | 0.828 |
| C1 | 0.720 | 0.791 | 0.845 | 0.751 | 0.817 | 0.899 | 0.956 | 0.898 | 0.907 | 0.953 | 0.772 | 0.926 | 0.819 | 0.815 | 0.830 |
| C2 | 0.607 | 0.659 | 0.647 | 0.697 | 0.716 | 0.805 | 0.858 | 0.791 | 0.850 | 0.881 | 0.796 | 0.814 | 0.822 | 0.829 | 0.865 |
| C3 | 0.675 | 0.683 | 0.715 | 0.704 | 0.669 | 0.777 | 0.776 | 0.833 | 0.778 | 0.790 | 0.788 | 0.779 | 0.826 | 0.799 | 0.776 |
| C4 | 0.795 | 0.781 | 0.785 | 0.758 | 0.755 | 0.832 | 0.861 | 0.850 | 0.852 | 0.837 | 0.853 | 0.881 | 0.846 | 0.863 | 0.868 |
| D | 0.796 | 0.785 | 0.790 | 0.782 | 0.787 | 0.863 | 0.870 | 0.888 | 0.864 | 0.859 | 0.848 | 0.850 | 0.865 | 0.846 | 0.847 |
| D1 | 0.907 | 0.922 | 0.956 | 0.923 | 0.942 | 0.918 | 0.925 | 0.942 | 0.900 | 0.922 | 0.928 | 0.933 | 0.962 | 0.949 | 0.944 |
| D2 | 0.573 | 0.624 | 0.569 | 0.528 | 0.537 | 0.690 | 0.696 | 0.679 | 0.689 | 0.626 | 0.666 | 0.608 | 0.591 | 0.547 | 0.546 |
| D3 | 0.830 | 0.841 | 0.861 | 0.876 | 0.903 | 0.962 | 0.974 | 0.953 | 0.957 | 0.954 | 0.937 | 0.946 | 0.939 | 0.943 | 0.955 |
| D4 | 0.765 | 0.785 | 0.758 | 0.771 | 0.733 | 0.818 | 0.898 | 0.909 | 0.919 | 0.865 | 0.829 | 0.855 | 0.854 | 0.896 | 0.882 |
| D5 | 0.807 | 0.838 | 0.839 | 0.802 | 0.805 | 0.921 | 0.936 | 0.951 | 0.892 | 0.900 | 0.911 | 0.915 | 0.943 | 0.900 | 0.917 |
| D6 | 0.752 | 0.742 | 0.766 | 0.775 | 0.752 | 0.793 | 0.804 | 0.846 | 0.819 | 0.826 | 0.794 | 0.801 | 0.822 | 0.864 | 0.869 |
| E | 0.761 | 0.762 | 0.782 | 0.784 | 0.776 | 0.820 | 0.832 | 0.851 | 0.859 | 0.855 | 0.826 | 0.823 | 0.833 | 0.845 | 0.843 |
| E1 | 0.737 | 0.775 | 0.680 | 0.699 | 0.709 | 0.816 | 0.824 | 0.687 | 0.782 | 0.811 | 0.841 | 0.840 | 0.739 | 0.705 | 0.782 |
| E2 | 0.721 | 0.740 | 0.731 | 0.760 | 0.747 | 0.863 | 0.879 | 0.871 | 0.900 | 0.887 | 0.792 | 0.820 | 0.804 | 0.785 | 0.797 |
| E3 | 0.737 | 0.776 | 0.776 | 0.757 | 0.750 | 0.805 | 0.842 | 0.798 | 0.802 | 0.758 | 0.825 | 0.814 | 0.815 | 0.846 | 0.834 |
| F | 0.754 | 0.761 | 0.758 | 0.812 | 0.806 | 0.892 | 0.915 | 0.901 | 0.914 | 0.914 | 0.876 | 0.880 | 0.881 | 0.933 | 0.920 |
| F1 | 0.759 | 0.762 | 0.819 | 0.795 | 0.825 | 0.890 | 0.901 | 0.868 | 0.922 | 0.937 | 0.835 | 0.879 | 0.862 | 0.918 | 0.922 |
| F2 | 0.848 | 0.871 | 0.877 | 0.830 | 0.828 | 0.940 | 0.967 | 0.985 | 0.961 | 0.941 | 0.869 | 0.916 | 0.972 | 0.940 | 0.930 |
| F3 | 0.798 | 0.784 | 0.785 | 0.816 | 0.810 | 0.890 | 0.870 | 0.892 | 0.923 | 0.924 | 0.888 | 0.877 | 0.843 | 0.921 | 0.899 |
| G | 0.653 | 0.665 | 0.565 | 0.658 | 0.683 | 0.772 | 0.769 | 0.697 | 0.775 | 0.799 | 0.727 | 0.692 | 0.627 | 0.752 | 0.794 |
| G1 | 0.715 | 0.687 | 0.476 | 0.629 | 0.672 | 0.595 | 0.695 | 0.797 | 0.879 | 0.792 | 0.580 | 0.592 | 0.695 | 0.885 | 0.814 |
| G2 | 0.636 | 0.653 | 0.571 | 0.663 | 0.648 | 0.747 | 0.760 | 0.691 | 0.772 | 0.780 | 0.719 | 0.693 | 0.611 | 0.736 | 0.736 |

**Table 7: The performance rank (1-28, 1 is best), measured by average ROC AUC, within a model and segmentation of each specific outcome. Outcomes are ordered by their average ranking across all models and segmentations. The labels of many outcomes are shortened, e.g., "D5 Confidence and self-esteem" becomes "Confidence". Abbreviations: func.: functioning, com.: communicate, MB: MentalBERT, ML: MentalLongformer, L2: Llama 2-7B, L3: Llama 3-8B.**

| Model
Label | Original
BERT | MB | ML | L2 | L3 | Monologue
BERT | MB | ML | L2 | L3 | Turns
BERT | MB | ML | L2 | L3 | Avg. rank |
|---|---|---|---|---|---|---|---|---|---|---|---|---|---|---|---|---|
| D3 Identity | 5 | 4 | 3 | 3 | 3 | 1 | 1 | 3 | 2 | 1 | 2 | 1 | 5 | 5 | 2 | 2.73 |
| F2 Hope | 4 | 2 | 2 | 6 | 4 | 2 | 2 | 1 | 1 | 4 | 7 | 6 | 1 | 6 | 6 | 3.60 |
| D1 Assertiveness | 1 | 1 | 1 | 1 | 1 | 4 | 6 | 5 | 10 | 10 | 3 | 3 | 2 | 4 | 4 | 3.73 |
| A3 Appetite | 10 | 16 | 18 | 9 | 12 | 5 | 3 | 2 | 8 | 3 | 1 | 2 | 3 | 1 | 1 | 6.27 |
| A4 Energy | 11 | 20 | 11 | 5 | 5 | 7 | 9 | 7 | 3 | 5 | 4 | 4 | 6 | 2 | 3 | 6.80 |
| D5 Confidence | 7 | 5 | 6 | 8 | 9 | 3 | 5 | 4 | 12 | 11 | 5 | 7 | 4 | 9 | 8 | 6.87 |
| F3 Optimism | 8 | 9 | 13 | 7 | 8 | 9 | 12 | 9 | 5 | 9 | 6 | 10 | 11 | 7 | 9 | 8.80 |
| F1 Peace of mind | 15 | 15 | 7 | 10 | 6 | 8 | 7 | 12 | 6 | 6 | 13 | 9 | 7 | 8 | 7 | 9.07 |
| C1 Global func. | 20 | 7 | 5 | 19 | 7 | 6 | 4 | 8 | 9 | 2 | 21 | 5 | 16 | 21 | 19 | 11.27 |
| D4 Processing past, present | 13 | 8 | 17 | 15 | 20 | 14 | 8 | 6 | 7 | 15 | 14 | 11 | 9 | 10 | 10 | 11.80 |
| B2 Resilience | 14 | 11 | 10 | 11 | 10 | 12 | 15 | 13 | 16 | 14 | 12 | 13 | 8 | 12 | 13 | 12.27 |
| C4 Social func. | 9 | 10 | 12 | 17 | 14 | 13 | 13 | 15 | 17 | 17 | 9 | 8 | 10 | 15 | 12 | 12.73 |
| A5 Self-harm | 3 | 3 | 4 | 2 | 2 | 21 | 26 | 28 | 13 | 8 | 27 | 21 | 26 | 3 | 5 | 12.80 |
| B3 Cognition | 22 | 17 | 21 | 12 | 19 | 11 | 10 | 10 | 4 | 7 | 22 | 20 | 19 | 14 | 16 | 14.93 |
| A7 Anxiety | 2 | 6 | 8 | 22 | 15 | 20 | 17 | 24 | 26 | 22 | 10 | 12 | 12 | 19 | 15 | 15.33 |
| A1 Mood and affect | 12 | 14 | 14 | 13 | 13 | 16 | 19 | 14 | 21 | 19 | 16 | 15 | 17 | 17 | 18 | 15.87 |
| E2 Family func. | 19 | 19 | 19 | 16 | 18 | 10 | 11 | 11 | 11 | 12 | 19 | 16 | 20 | 25 | 22 | 16.53 |
| A6 Suicidality | 6 | 21 | 9 | 4 | 11 | 17 | 25 | 23 | 15 | 21 | 8 | 23 | 21 | 23 | 26 | 16.87 |
| D6 Feeling seen | 16 | 18 | 16 | 14 | 16 | 23 | 20 | 16 | 22 | 20 | 18 | 19 | 14 | 13 | 11 | 17.07 |
| E3 Friendships | 17 | 12 | 15 | 18 | 17 | 19 | 16 | 18 | 23 | 27 | 15 | 17 | 18 | 16 | 17 | 17.67 |
| C2 Executive func. | 26 | 24 | 24 | 23 | 21 | 18 | 14 | 21 | 19 | 13 | 17 | 18 | 15 | 18 | 14 | 19.00 |
| E1 Com. feelings, thoughts | 18 | 13 | 22 | 21 | 22 | 15 | 18 | 26 | 24 | 23 | 11 | 14 | 23 | 27 | 24 | 20.07 |
| C3 Academic func. | 23 | 23 | 20 | 20 | 25 | 24 | 21 | 17 | 25 | 25 | 20 | 22 | 13 | 22 | 25 | 21.67 |
| A2 Anger | 28 | 28 | 23 | 24 | 27 | 25 | 23 | 20 | 18 | 16 | 25 | 24 | 22 | 20 | 23 | 23.07 |
| G1 Parental wellbeing | 21 | 22 | 28 | 27 | 24 | 28 | 28 | 19 | 14 | 24 | 28 | 28 | 25 | 11 | 21 | 23.20 |
| B1 Behavioural activation | 25 | 27 | 25 | 25 | 23 | 22 | 24 | 22 | 20 | 18 | 24 | 25 | 24 | 24 | 20 | 23.20 |
| G2 Parental support | 24 | 25 | 26 | 26 | 26 | 26 | 22 | 25 | 27 | 26 | 23 | 26 | 27 | 26 | 27 | 25.47 |
| D2 Autonomy | 27 | 26 | 27 | 28 | 28 | 27 | 27 | 27 | 28 | 28 | 26 | 27 | 28 | 28 | 28 | 27.33 |