# OpenReview forum: "Using Large Language Models to Detect Outcomes of Qualitative Studies on Adolescent Depression"
_KDD.org/2024/Workshop/AIDSH — KDD-AIDSH 2024 Poster_

### Official Review · Reviewer_hoik · 2024-06-09
**LLM for mental health on adolescent depression, but only in outcomes of detection**

**Rating:** 5
**Confidence:** 2

**Review:**

Summary:
In this paper, authors present a novel approach to use LLM to detect outcomes for mental health, especially on adolescent depression. Thus, they use the embeddings to classify the high-level domains from their qualitative analysis.

Strengths:
This article provides an interesting perspective to use the LLM to process the text on adolescent depression, and it is demonstrated to be successful on a range of classification tasks.  It is a solid work with complete experiments.

Weaknesses:
In general, the technical part of this article is relatively weak in innovation. It only uses language models to do some basic text classification tasks, just like traditional NLP tasks.

In addition, the article only uses the most advanced LLM embedding for classification tasks, which may limit its performance to some extent. Because LLM, such as Llama 3, their training method and architecture (decoder-only) are not suitable for some text understanding tasks, even though they do well in text understanding. The author should try other classification tasks to test the performance of LLM as much as possible. For example, zero-shot classification, few-shot classification tasks, etc. These are the most suitable scenarios for LLM.

As mentioned above, I would mainly ask you to add some new tasks for LLM. I think It is easy to do that.

---

### Official Review · Reviewer_ZoH7 · 2024-06-17
**Review of "Using Large Language Models to Detect Outcomes in Qualitative Studies on Adolescent Depression"**

**Rating:** 4
**Confidence:** 4

**Review:**

This manuscript attempts to leverage Large Language Models (LLMs) for extracting features related to depression from clinical data, presenting a novel approach. The use of detailed and extensive tables to present results provides clarity and enriches the content. However, there are several areas where the manuscript could be significantly improved.

- Justification for Data Choice: The manuscript states that the use of clinical data over social media data, but I'm wondering why this choice is superior for depression identification.
- Outdated Literature Review: The related work section is mostly about traditional machine learning and deep learning methods that are dated. This section needs substantial updates to include more recent advancements in NLP and specifically LLMs for depression detection.
- Lack of Comparative Analysis: The experimental results are presented only as absolute AUC values without any comparison to baseline methods, which undermines the persuasiveness of the results. Mentioning and comparing to at least some baseline methods in the related work would enhance the credibility.
- Redundancy in Classification Discussion: The section on using LR for classification seems redundant. The main focus of the manuscript is on using LLMs to extract certain features of depression, having already validated the AUC for each indicator and high-level domain AUC. If the intention is to further demonstrate that LLM-extracted indicators are beneficial for depression classification, a comparison should be introduced. Specifically, a baseline should be introduced to show that features extracted by other methods are less accurate in classification when using LR compared to features extracted by LLMs.
- Figure Clarity: Figure 1 is described as too vague and overloaded with information, making it difficult to grasp the main points. A suggestion is made to replace this figure with tables from the appendix that focus on AUC values for a clearer and more detailed presentation.
- Small Dataset: The dataset size is relatively small, with only 34 cases, which may not provide enough variability to generalize the findings effectively.

---

### Decision · Program_Chairs · 2024-06-28

Accept (Poster)